

# Phylogeny of Libellulidae (Odonata: Anisoptera): comparison of molecular and morphology-based phylogenies based on wing morphology and migration

Shu-Ting Huang[1], Hai-Rui Wang[2], Wan-Qin Yang[1], Ya-Chu Si[1], Yu-Tian Wang[1], Meng-Lian Sun[1], Xin Qi[1] and Yi Bai[1]

[1] Zhejiang Provincial Key Laboratory of Plant Evolutionary Ecology and Conservation, Taizhou University, Taizhou, Zhejiang, China
[2] Sports Science Institute, Taizhou University, Taizhou, Zhejiang, China

Corresponding author
Yi Bai, baiyi@tzc.edu.cn

## ABSTRACT

**Background**. Establishing the species limits and resolving phylogenetic relationships are primary goals of taxonomists and evolutionary biologists. At present, a controversial question is about interspecific phylogenetic information in morphological features. Are the interspecific relationships established based on genetic information consistent with the traditional classification system? To address these problems, this study analyzed the wing shape structure of 10 species of Libellulidae, explored the relationship between wing shape and dragonfly behavior and living habits, and established an interspecific morphological relationship tree based on wing shape data. By analyzing the sequences of mitochondrial *COI* gene and the nuclear genes *18S*, *28S rRNA* and *ITS* in 10 species of dragonflies, the interspecific relationship was established.

**Method**. The wing shape information of the male forewings and hindwings was obtained by the geometric morphometrics method. The inter-species wing shape relationship was obtained by principal component analysis (PCA) in MorphoJ1.06 software. The inter-species wing shape relationship tree was obtained by cluster analysis (UPGMA) using Mesquite 3.2 software. The *COI*, *18S*, *ITS* and *28S* genes of 10 species dragonfly were blasted and processed by BioEdit v6 software. The Maximum Likelihood(ML) tree was established by raxmlGUI1.5b2 software. The Bayes inference (BI) tree was established by MrBayes 3.2.6 in Geneious software.

**Results**. The main difference in forewings among the 10 species of dragonfly was the apical, radial and discoidal regions dominated by the wing nodus. In contrast, the main difference among the hindwings was the apical and anal regions dominated by the wing nodus. The change in wing shape was closely related to the ability of dragonfly to migrate. The interspecific relationship based on molecular data showed that the species of *Orthetrum* genus branched independently of the other species. Compared to the molecular tree of 10 species, the wing shape clustering showed some phylogenetic information on the forewing shape (with large differences on the forewing shape tree vs. molecular tree), and there was no interspecific phylogenetic information of the hindwing shape tree vs. molecular tree.

**Conclusion**. The dragonfly wing shape characteristics are closely related to its migration ability. Species with strong ability to migrate have the forewing shape that is longer and narrower, and have larger anal region, whereas the species that prefer short-distance

hovering or standing still for a long time have forewing that are wider and shorter, and the anal region is smaller. Integrating morphological and molecular data to evaluate the relationship among dragonfly species shows there is some interspecific phylogenetic information in the forewing shape and none in the hindwing shape. The forewing and hindwing of dragonflies exhibit an inconsistent pattern of morphological changes in different species.

## INTRODUCTION

The morphological evolution of insects and the formation of species have been the scientific issues that taxonomists, evolutionary biologists and ecologists are constantly exploring (*Misof et al., 2014*; *Crispo, 2008a*; *Crispo, 2008b*). In natural selection and adaptation, insects have formed diverse phenotypic characteristics and genetic structure (*Lundsgaard-Hansen, Matthews & Seehausen, 2014*; *Schneider, 2000*). With the continuous development and improvement of modern molecular biology technology, establishing reliable inter-species ancestry from a genetic perspective has been well documented (*Mack & Nachman, 2017*; *Soria-Carrasco et al., 2014*; *Gompert et al., 2014*). However, the traditional classification system is based on the morphological characteristics of the species. Currently, the hot issue to be explored is whether the interspecific relationships established by morphological features can be supported by the molecular data or, in other words, to what extent the current classification system is supported (*Virgilio et al., 2010*; *Lukhtanov, Sourakov & Zakharov, 2016*; *Renaud, Savage & Adamowicz, 2012*). In the last decade, the traditional views have been that morphological characteristics that involve ecological adaptation and behavioral problems (such as living in the same ecological niche and having similar feeding behaviors and patterns of movement) lead to morphological similarities (*Stern & Orgogozo, 2008*).The genetic diversity may not necessarily be related to morphological differences (*Robertson, 1959*; *Wlikens, 1971*). However, in recent decades, studies in molecular biology and developmental biology have suggested that mutations in gene expression regulation may promote phenotypic evolution, especially the change in morphological characteristics (*Kaessmann, 2010*; *Rabosky, 2012*; *Crispo, 2008a*; *Crispo, 2008b*). It indicates that the differences in the genetic structure are predictable, and to a certain extent, they will result in differences in the morphological structure. These contradictory views are common in evolution of the related species of insects (*Víctor & Zúñiga, 2016*; *Heikkilä et al., 2015*)). In recent years, with the constant development and improvement of morphological measurement technology, the geometric morphometrics represents a powerful tool to investigate further the shape evolution in systematics (*Klingenberg & Marugán-Lobón, 2013*). Geometric morphometrics is an advanced method of morphological analysis in biology; based on the curve, landmark point and semi-landmark point data of the homologous locus concept, it can accurately quantify the phenotypic traits of organisms

and explore the morphological evolution of populations (*Cooke & Terhune, 2015*; *Baylac, Villemant & Simbolotti, 2003*).

A large number of studies have shown that the morphology-based interspecific relationship is basically consistent with the interspecific relationship established by molecular data when the morphological characteristics are selected judiciously (*Grzywacz et al., 2017*; *Noguerales, Cordero & Ortego, 2018*). *Marín et al. (2017)* showed that the interspecific relationship of Nymphalidae based on wing shape and wing vein was consistent with that based on molecular data. It indicated the existence of phylogenetic information in the insect wing morphology. *Francischini et al. (2017)* used *COI* gene and female genital structure to illustrate the interspecific relationship of *Diatraea* by the molecular and morphological methods, and the results obtained by the two methods were consistent in the classification of the interspecific relationships. Similar studies were conducted by *Ortego, Aguirre & Cordero (2012)* on the population differentiation of *Mioscirtus wagneri* locust in different geographical regions by using quantitative morphological features (anterior and posterior plates) and mtDNA, suggesting that morphology-based geographical differentiation correlated with geographical differentiation at the molecular level. Therefore, the morphological data and molecular data can support the interrelationships of the major taxonomic groups in animals. However, the establishment of interspecific phylogenetic relationships based on morphology and molecular data can also lead to inconsistent results. For example, *Bocek & Bocak (2017)* showed the morphology of beetle pronotum cannot fully support interspecific phylogenetic relationships. *Bapst, Schreiber & Carlson (2018)* used molecular and morphological data to study the interspecific relationship of *Branchiopoda* and found that the morphological data did not have interspecific phylogenetic information. Due to the common phenomenon of convergent evolution of morphological and ecological traits in nature, the morphological traits may only reveal some differences in phenotype of different research objects, but to accurately reflect the phylogenetic relationships among species, the morphological data need to be combined with the molecular data for synthesis.

This research selected 10 species of dragonfly from the same habitat to study their interspecific relationships. Libellulidae belong to Odonata, and have two pairs of large and transparent membranous wings, with the wing veins clearly visible; the shape and direction of the wing veins are often used as an important classification basis for dragonflies (*Outomuro, Dijkstra & Johansson, 2013*; *Fauziyah et al., 2014*; *Rajabi et al., 2016*). Geometric morphometrics was used to analyze the morphological differences among species. The mitochondrial gene *COI* and nuclear genes *18S*, *ITS* and *28S* were used to analyze the phylogenetic relationships among the species. We analyzed the phylogenetic relationships based on wing-type features as well as on molecular data. Accordingly, this study addressed the following questions: (1) what was the relationship between the characteristics of the wing shape and the behavior of the dragonfly? (2) Did the wing shape and wing vein contain interspecific phylogenetic information? (3) Did forewings and hindwings exhibit a consistent pattern of morphological changes in different species?

**Table 1  The species name, genus, subfamily, family and number of 10 species of Libellulidae.**

| Species | Genus | Subfamily | Family | Numbers |
|---|---|---|---|---|
| *Orthetrum albistylum* (Selys) | *Orthetrum* | Libellulinae | Libellulidae | 8 |
| *Orthetrum melania* (Selys) | *Orthetrum* | Libellulinae | Libellulidae | 5 |
| *Orthetrum testaceus* (Burmeister) | *Orthetrum* | Libellulinae | Libellulidae | 6 |
| *Acisoma panorpoides* (Rambur) | *Acisoma* | Sympetrinae | Libellulidae | 9 |
| *Deielia phaon* (Selys) | *Deielia* | Sympetrinae | Libellulidae | 15 |
| *Crocothemis servilia* (Drury) | *Crocothemis* | Sympetrinae | Libellulidae | 6 |
| *Trithemis aurora* (Burmeister) | *Trithemis* | Trithemistinae | Libellulidae | 13 |
| *Pseudothemis zonata* (Burmeister) | *Pseudothemis* | Trithemistinae | Libellulidae | 3 |
| *Tramea virginia* (Rambur) | *Tramea* | Trameinae | Libellulidae | 7 |
| *Pantala flavescens* (Fabricius) | *Pantala* | Trameinae | Libellulidae | 9 |
| *Anotogaster sieboldii* (Selys) | *Anotogaster* | | Cordulegastridae | 3 |

## MATERIALS & METHODS

### Materials and data acquisition
#### Specimen collection and image acquisition
In order to explore the relationship between the wing shape of different species of dragonfly in Odonata, we collected dragonflies from May to October 2018 at Xikou reservoir (28°48′N, 121°25′E) and surrounding areas in Taizhou City. After classification and identification, the males of 10 species of dragonfly were selected for this research. A total of 84 individuals were studied. The species names and numbers are shown in Table 1. The wings of all specimens were spread; then, the left forewing and the left hindwing were taken and pressed between two slides to make slide specimens. The forewings and hindwings were photographed with a Nikon 5100 camera, fixed on a stand. A ruler and a slide specimen were placed on the same horizontal plane for photographing. All photographs were made using the identical camera settings and were saved in a picture format for later use.

#### Landmark data acquisition
The TPSdig2 software (*Rohlf, 2006*) was used to digitize wing images of 10 species of Libellulidae, identifying 26 landmarks on the forewing and 27 on the hindwing (in each case, including two on a ruler) (Fig. 1). The landmark-based geometric morphometrics method was applied to study the morphological diversity in wing size and shape. We set landmarks at the intersections of wing veins with the wing margin and intersections of cross veins with major veins and vein branch points (Table 2), which was according to *Rohlf & Corti (2000)*.

#### Wing shape analysis
The forewing and hindwing shape information was input into CoordGen software (*Rohlf & Slice, 1990*) in the IMP series package. Based on the ruler, the errors caused by the focal length of the photograph were eliminated, and the datum line was set. To examine wing-shape variation, digitized landmark data were subjected to generalized procrustes superimposition to standardize the size of the landmark configurations and

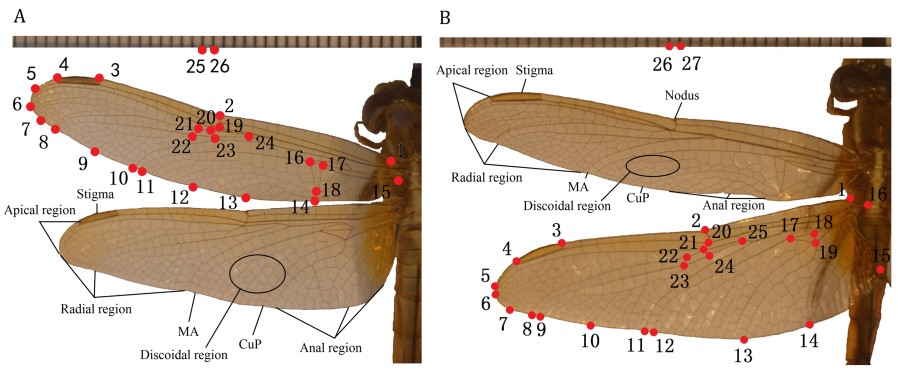

**Figure 1** **Landmarks on the forewing and hindwing of Libellulidae.** (A) Landmarks 1 to 24 on forewing, 25 to 26 are rulers; (B) Landmarks 1 to 25 on hindwing, 26 to 27 are rulers.

**Table 2** **Definition and numbering of the landmarks.**

| L. of forewing | Definition | L. of hindwing | Definition |
| --- | --- | --- | --- |
| 1 | Initial of costa | 1 | Initial of costa |
| 2 | Nodus | 2 | Nodus |
| 3 | Left of stigma | 3 | Left of stigma |
| 4 | Right of stigma | 4 | Right of stigma |
| 5 | Midpoint of 4 and 6 | 5 | End of sub-costa |
| 6 | End of RP1 | 6 | End of RP1 |
| 7 | Midpoint of 6 and 8 | 7 | Midpoint of 6 and 8 |
| 8 | End of RP2 | 8 | End of RP2 |
| 9 | Midpoint of 8 and 10 | 9 | End of IRP2 |
| 10 | End of RP3-4 | 10 | Midpoint of 9 and 10 |
| 11 | End of MA | 11 | End of RP3-4 |
| 12–14 | Anal region | 12 | End of MA |
| 15 | End of the anal vein | 13 | End of CuP |
| 16–18 | triangle region | 14–15 | Anal region |
| 19–20 | Sub-nodu | 16 | End of the anal vein |
| 21–24 | Midvein region | 17–19 | triangle region |
| | | 20–21 | Sub-nodu |
| | | 22–25 | Midvein region |

eliminate differences caused by translation and rotation (*Adams, Rohlf & Slice, 2004*). All standardized data were converted into a two-dimensional data format.

## Methods and analysis
### Statistical analysis of morphological data

The standardized morphological information data were imported into MorphoJ1.06d software (*Klingenberg, 2009*), and the morphological changes of 10 species of dragonfly were analyzed by principal component analysis (PCA), Procrustes analysis and Discriminant analysis. The first two main components were extracted as scatter plots of forewings and

hindwings. In the MorphoJ1.06d software, the thin-plate spline analysis (*Bookstein, 1989*) was performed, and the difference in landmark points was analyzed. The visualized legend was used to show the variation in forewings and hindwings in the first two principal components.

### Acquisition of molecular data

The DNA barcode data of 10 species of dragonfly was obtained from the NCBI website. We obtained *Cytochrome oxidase subunit I* (*COI*) gene of each species with length of 349 bp and *18S rRNA*, *Internal Transcribed Space1* (*ITS1*) + *5.8s rRNA* + *Internal Transcribed Space2* (*ITS2*) and *28S rRNA* of each species with length of 747 bp. All data were imported into BioEdit v6 software for editing, and the built-in clustalw was used to blast sequences (*Hall, 1999*). Total obtained *COI + 18s +ITS +28s* gene data with a length of 1,096 bp was used to construct the Maximum likelihood (ML) tree and the Bayesian inference (BI) tree. The gene sequence numbers and related information are shown in Table 3.

### Establishment of morphological and molecular phylogenetic trees

In this study, Mesquite 3.2 software (*Maddison & Maddison, 2018*) was used to cluster the morphological characteristics of forewings and hindwings of 10 dragonfly species. The cluster analysis was based on the landmark data for forewings and hindwings of each species established as a matrix. The distance among the taxa represented uncorrected distance. Then, the relationships among the populations were further summarized based on the unweighted pair-group method with arithmetic averages (UPGMA) to build forewing and hindwing shape trees (*Ramírez-Sánchez, Luna & Cramer, 2016*).

The mitochondrial gene *COI* and nuclear gene *18S +ITS +28S* were analyzed using ML and BI, respectively. Best-fit evolutionary models were determined with MEGA6 software (*Tamura et al., 2013*); we obtained the GTR+G+I model with the lowest Bayesian Information Criterion score, suggesting it was the best-fit model. The ML analysis was done using raxmlGUI1.5b2 (*Silvestro & Michalak, 2012*), under a GTRGAMMAI model. The Bootstrap supports and trees were obtained by running rapid bootstrap analysis of 1,000 replicates followed by a search for the best scoring ML tree. The BI was done using MrBayes 3.2.6 (*Huelsenbeck & Ronquist, 2001*) executed from within Geneious (*Kearse et al., 2012*). The model used the above-mentioned GTR+G+I model (*Yang, 1994*). All BI analyses consisted of 1. $1 \times 10^6$ generations of Markov Chain Monte Carlo searches containing 4 chains, heated chain temperature of 0.2 and burn-in of 100,000 generations. Compound Dirichlet priors for branch lengths were assigned to avoid branch-length overestimation using the following: prset brlenspr = unconstrained, gammadir (1.0,0.1,1.0,1.0) shapepr = exponential(10.0). Trees were saved every 1,000 generations. The confidence values of the BI tree were presented as the Bayesian posterior probabilities (BPP) in percentages. Phylogenetic analyses were performed for each of the gene sequences.

We using Dendroscope 3.6.2 software (*Huson et al., 2007*) to edit the phylogenetic trees and get tanglegram between phylogeny constructed using morphological data and that constructed using molecular data.

**Table 3  The famliy, subfamily and genBank number of 10 species Libellulidae.**

| Family | Subfamily | Species | GenBank number | |
| --- | --- | --- | --- | --- |
| | | | Mitochondria COI | Nuclear 18S rRNA+ITs1+5.8s+ITs2+28S rRNA |
| Libellulidae | Libellulinae | *Orthetrum albistylum* | MF358741.1 | LC366177.1 |
| Libellulidae | Libellulinae | *Orthetrum albistylum* | MF358740.1 | AB781474.1 |
| Libellulidae | Libellulinae | *Orthetrum albistylum* | MF358739.1 | AB781473.1 |
| Libellulidae | Libellulinae | *Orthetrum melania* | LC099937.1 | LC099933.1 |
| Libellulidae | Libellulinae | *Orthetrum melania* | AB709043.1 | AB707165.1 |
| Libellulidae | Libellulinae | *Orthetrum melania* | AB709085.1 | AB707187.1 |
| Libellulidae | Libellulinae | *Orthetrum testaceus* | KU496907.1 | KJ802972.1 |
| Libellulidae | Libellulinae | *Orthetrum testaceus* | KU496905.1 | KJ802970.1 |
| Libellulidae | Libellulinae | *Orthetrum testaceus* | MF774527.1 | KJ802969.1 |
| Libellulidae | Sympetrinae | *Acisoma panorpoides* | KX281827.1 | AB707046.1 |
| Libellulidae | Sympetrinae | *Acisoma panorpoides* | KX281825.1 | AB707045.1 |
| Libellulidae | Sympetrinae | *Acisoma panorpoides* | KX281824.1 | FN356030.1 |
| Libellulidae | Sympetrinae | *Deielia phaon* | AB708961.1 | AB707069.1 |
| Libellulidae | Sympetrinae | *Deielia phaon* | AB708962.1 | AB707068.1 |
| Libellulidae | Sympetrinae | *Deielia phaon* | AB708963.1 | AB707066.1 |
| Libellulidae | Sympetrinae | *Crocothemis servilia* | JN119571.1 | LC366268.1 |
| Libellulidae | Sympetrinae | *Crocothemis servilia* | MF774561.1 | LC366266.1 |
| Libellulidae | Sympetrinae | *Crocothemis servilia* | MF774554.1 | LC366265.1 |
| Libellulidae | Trithemistinae | *Trithemis aurora* | MF358792.1 | AB707343.1 |
| Libellulidae | Trithemistinae | *Trithemis aurora* | MF358785.1 | AB707342.1 |
| Libellulidae | Trithemistinae | *Trithemis aurora* | MF358776.1 | GU323038.1 |
| Libellulidae | Trithemistinae | *Pseudothemis zonata* | MF358738.1 | AB707212.1 |
| Libellulidae | Trithemistinae | *Pseudothemis zonata* | KF257079.1 | AB707212.1 |
| Libellulidae | Trameinae | *Tramea virginia* | AB709228.1 | AB707335.1 |
| Libellulidae | Trameinae | *Tramea virginia* | AB709225.1 | AB707331.1 |
| Libellulidae | Trameinae | *Tramea virginia* | AB709227.1 | AB707332.1 |
| Libellulidae | Trameinae | *Pantala flavescens* | KR080133.1 | LC366168.1 |
| Libellulidae | Trameinae | *Pantala flavescens* | KR080114.1 | AB707211.1 |
| Libellulidae | Trameinae | *Pantala flavescens* | KR080079.1 | LC366076.1 |
| Cordulegastridae | | *Anotogaster sieboldii* | EF155476.1 | AB706931.1 |
| Cordulegastridae | | *Anotogaster sieboldii* | EF155431.1 | AB706930.1 |

# RESULTS

## Morphological differences in forewings and hindwings of 10 species of dragonflies (Libellulidae)

The wing shape data were analyzed by PCA and centroid size to find out the shape variation (Fig. 2A). The first two PCs accounted for 35.09% and 21.77% of the variation, with the cumulative variation explaining 56.86% of the total shape variation in forewings. Procrustes analysis (Table 4) of forewings showed *Deielia phaon* and *Pantala flavescens* to have the smallest distance (0.006), suggesting their forewing shape differences was
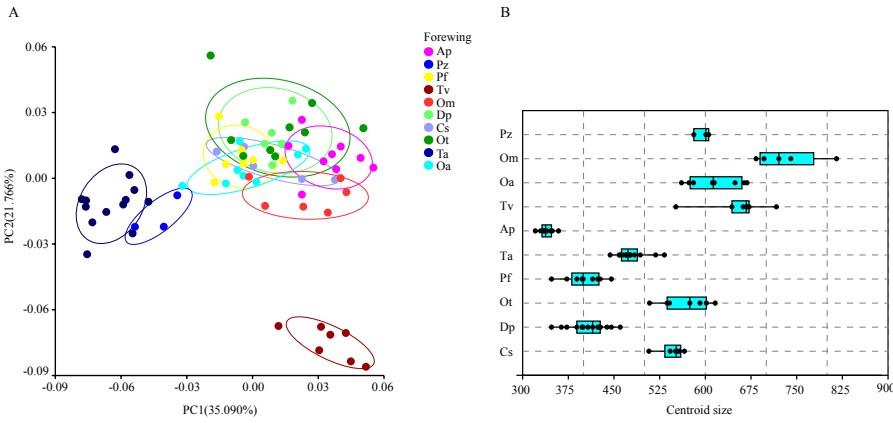

**Figure 2** PCA (A) and Centroid size analysis (B) of forewings of 10 dragonfly species (Libellulidae). Ap, *Acisoma panorpoides*; Pz, *Pseudothemis zonata*; Pf, *Pantala flavescens*; Tv, *Tramea virginia*; Om, *Orthetrum melania*; Dp, *Deielia phaon*; Cs, *Crocothemis servilia*; Ot, *Orthetrum testaceus*; Ta, *Trithemis aurora*; Oa, *Orthetrum albistylum*.

small. *Trithemis aurora* and *Tramea virginia* had the largest distance (0.120), meaning their forewing shape differences were large. Discriminant analysis results showed no significant difference in forewing shapes between *Deielia phaon* and *Pantala flavescens* ($P = 1.000$), significant differences between *Crocothemis servilia* and *Orthetrum albistylum* ($P = 0.023$) *and Crocothemis servilia* and *Orthetrum melania* ($P = 0.042$), and strongly significant differences among the other species ($P < 0.01$). A scatter plot (Fig. 2A) of the first and second principal components showed that on the PC1 axis, *Trithemis aurora*, *Pseudothemis zonata* and *Orthetrum testaceus* were mainly distributed on the negative direction, whereas the other seven species were mainly distributed on the positive direction. Taking into account the profile plots of the wing veins (Fig. 3), the differences mainly occurred in the apical region (LM6-8) and the discoidal region (LM11-14). On the PC2 axis, *Orthetrum melania*, *Tramea virginia*, *Trithemis aurora* and *Pseudothemis zonata* were positioned mainly on the negative direction, and the other six dragonfly species were distributed mainly on the positive direction. The forewing profiles (Fig. 3) showed that the differences occurred mainly in the apical (LM6-8) and the radial region (LM8-10). Centroid Size Analysis (Fig. 2B) results showed that *Deielia phaon*, *Pantala flavescens* and *Acisoma panorpoides* had smaller, and *Tramea virginia* and *Orthetrum melania* had larger forewings.

The hindwing shape data were analyzed via PCA and Centroid size to find out the shape variation (Fig. 4A). The first two PCs accounted for 37.08% and 21.41% of the variation, with the cumulative variation explaining 58.49% of the total shape variation in hindwings. Procrustes analysis (Table 4) on hindwings showed *Crocothemis servilia and Orthetrum testaceus* with the smallest distance (0.026), suggesting their hindwing shapes were similar. The *Acisoma panorpoides* and *Tramea virginia* had the largest distance (0.132), indicating relatively large differences in their hindwing shapes. Discriminant analysis showed no significant difference in hindwing shapes between *Orthetrum melania* and

**Table 4  The procrustes distance of forewing and hindwing shape among 10 species of Libellulidae.**

| Hindwing shape distance Forewing shape distance | Cs | Dp | Ot | Pf | Ta | Ap | Tv | Oa | Om | Pz |
|---|---|---|---|---|---|---|---|---|---|---|
| Cs |  | 0.059** | **0.026** | 0.083** | 0.054** | 0.079** | 0.107** | 0.034 | 0.045 | 0.055 |
| Dp | 0.042** |  | 0.051** | 0.096** | 0.074** | 0.044** | 0.124** | 0.051** | 0.073** | 0.074* |
| Ot | 0.034* | 0.041** |  | 0.083** | 0.055** | 0.064** | 0.106** | 0.040** | 0.051* | 0.056** |
| Pf | 0.041** | **0.006** | 0.042** |  | 0.068** | 0.116** | 0.118** | 0.073** | 0.063** | 0.051** |
| Ta | 0.082** | 0.084** | 0.069** | 0.085** |  | 0.090** | 0.093** | 0.062** | 0.049** | 0.041** |
| Ap | 0.066** | 0.045** | 0.060** | 0.046** | 0.108** |  | 0.132** | 0.078** | 0.095** | 0.096* |
| Tv | 0.094** | 0.101** | 0.099* | 0.102** | 0.120** | 0.091** |  | 0.116** | 0.086** | 0.112* |
| Oa | 0.032 | 0.041** | 0.033** | 0.040* | 0.071** | 0.064** | 0.094** |  | 0.048** | 0.054** |
| Om | 0.033 | 0.059** | 0.056* | 0.058* | 0.096** | 0.076** | 0.084** | 0.046* |  | **0.041** |
| Pz | 0.094** | 0.088** | 0.089** | 0.090* | 0.072* | 0.106** | 0.116* | 0.082* | 0.108* |  |

**Notes.**

*represent significance level <0.01

**represent significance level <0.001.

Ap, *Acisoma panorpoides*; Pz, *Pseudothemis zonata*; Pf, *Pantala flavescens*; Tv, *Tramea virginia*; Om, *Orthetrum melania*; Dp, *Deielia phaon*; Cs, *Crocothemis servilia*; Ot, *Orthetrum testaceus*; Ta, *Trithemis aurora*; Oa, *Orthetrum albistylum*.

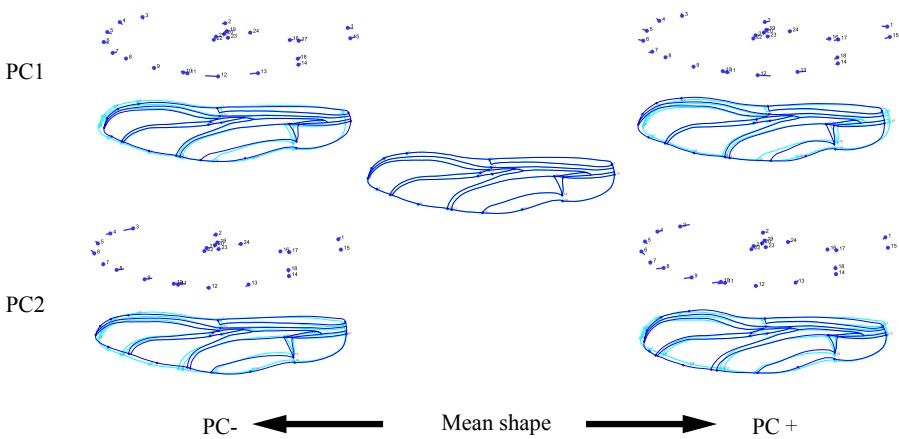

PC1

PC2

PC-  ←  Mean shape  →  PC +

**Figure 3  Thin-plate spline analysis of forewing profiles of 10 dragonfly species (Libellulidae).** Each profile represents the deformations in wing shape in extreme conditions for each PC.

*Pseudothemis zonata* ($P = 0.111$), significant differences between *Crocothemis servilia* and *Orthetrum testaceus* ($P = 0.034$), *Crocothemis servilia* and *Orthetrum albistylum* ($P = 0.046$) and *Crocothemis servilia* and *Orthetrum melania* ($P = 0.014$), and strongly significant differences between *Crocothemis servilia* and *Pseudothemis zonata* ($P = 0.001$) and among the other species ($P < 0.01$). A scatter plot of PC 1 vs. PC2 (Fig. 4A) showed that on the PC1 axis, *Orthetrum testaceus*, *Orthetrum melania*, *Crocothemis servilia*, *Deielia phaon*, and *Acisoma panorpoides* were positioned mainly on the positive direction, and the other five dragonfly species were distributed mainly on the negative direction. Taking into account

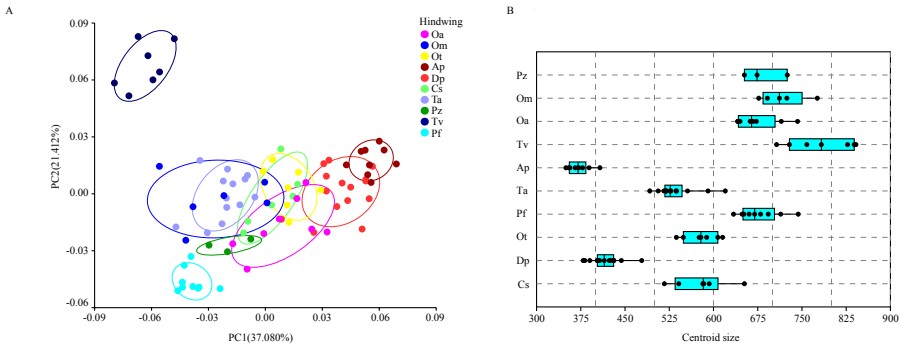

**Figure 4** **PCA (A) and Centroid sze analysis (B) of hindwings of 10 dragonfly species (Libellulidae).**
Ap, *Acisoma panorpoides*; Pz, *Pseudothemis zonata*; Pf, *Pantala flavescens*; Tv, *Tramea virginia*; Om, *Orthetrum melania*; Dp, *Deielia phaon*; Cs, *Crocothemis servilia*; Ot, *Orthetrum testaceus*; Ta, *Trithemis aurora*; Oa, *Orthetrum albistylum*.

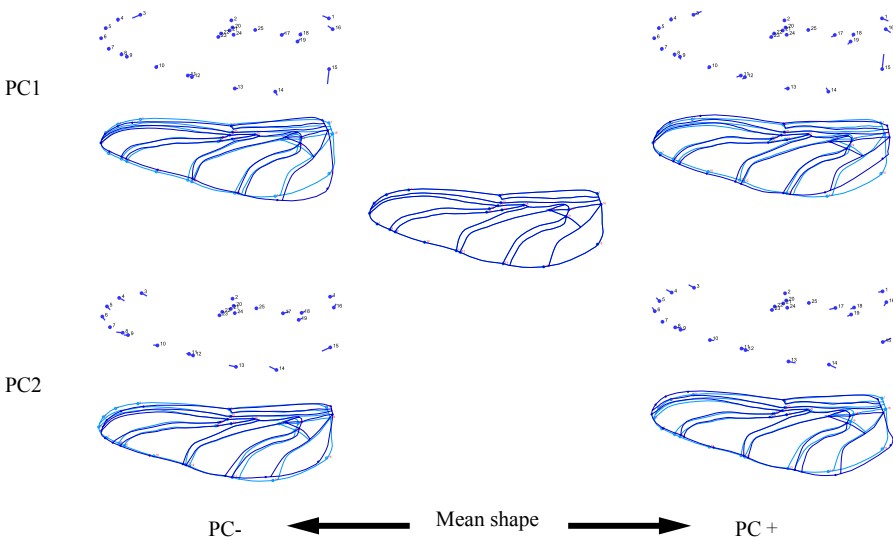

**Figure 5** **Thin-plate spline analysis of hindwing profiles of 10 dragonfly species (Libellulidae).** Each
profile represents the deformations in wing shape in extreme conditions for each PC.

the profile plot of the wing vein (Fig. 5), the differences in hindwings occurred mainly
in the anal region (LM13-16). On the PC2 axis, *Tramea virginia*, *Acisoma panorpoides*,
*Orthetrum testaceus,* and *Deielia phaon* were distributed mainly on the positive direction,
and the other six species were positioned mainly on the negative direction. The profiles of
hindwing veins (Fig. 5) showed that the differences occurred mainly in the apical (LM6-8)
and the anal region (LM13-16). Centroid Size Analysis (Fig. 4B) showed that *Deielia phaon*
and *Acisoma panorpoides* had smaller hindwings, whereas *Tramea virginia and Orthetrum
melania* had larger hindwings.

Combining the results of the two analyses (PCA and Centroid size), the change rules
of forewing shape among species were different to those of hindwing shape. For example,
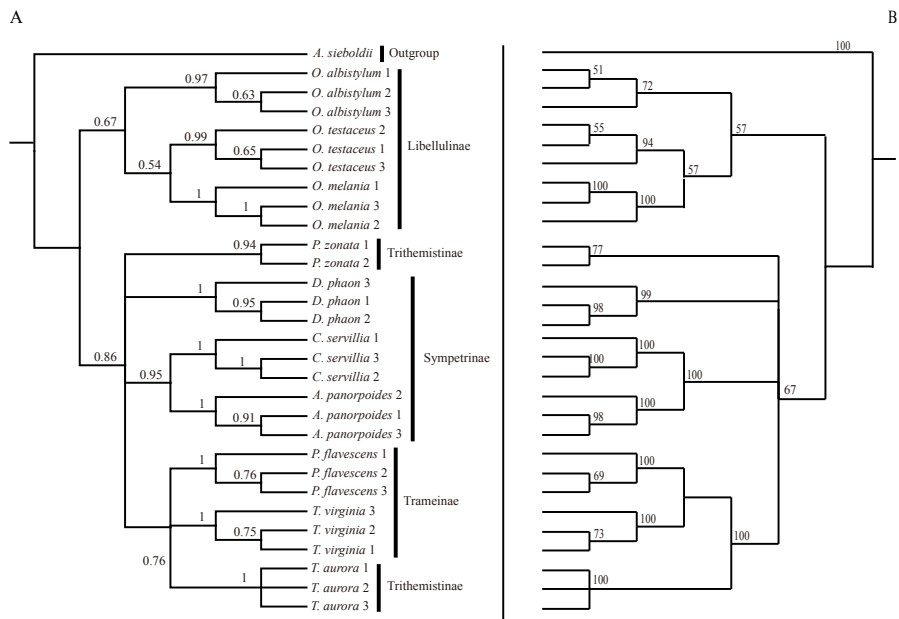

**Figure 6  Bayesian Inference tree (A) and maximum likelihood tree (B).** The phylogenetic trees were constructed based on molecular data of the mitochondrial *COI* and nuclear *18S rRNA + ITS1 + 5.8S rRNA + ITS2 + 28S rRNA* genes.

the forewing and hindwing shape analysis of *Trithemis aurora* showed large differences on the PC1 axis. In Centroid size analysis, *Orthetrum melania* had the biggest forewings, but *Tramea virginia* had the biggest hindwings.

## Analysis of interspecific relationships based on molecular data

Analysis of the interspecific relationship among 10 species of dragonfly by the ML method (Fig. 6) divided them into two main branches, with *Orthetrum* species (subfamily Libellulinae) in one branch, having a distant relationship with other species. The remaining seven species were divided into four branches, forming a paraphyletic group. *Deielia phaon* was on a separate branch, whereas *Acisoma panorpoides* and *Crocothemis servilia* were clustered into a branch with a high degree of support (all three species belonging to subfamily Sympetrinae). *Pseudothemis zonata* was on a separate branch (subfamily Trithemistinae). *Pantala flavescens*, *Tramea virginia* and *Trithemis aurora* were clustered into a branch with a high degree of support (*Pantala flavescens* and *Tramea virginia* belonging to subfamily Trameinae, and *Trithemis aurora* to subfamily Trithemistinae).

The phylogenetic tree obtained by the BI method was basically consistent with the relationship tree obtained by the ML method. Although the BI tree divided further the relationship among the seven species in the four paraphyletic groups, the support was not high, so the interspecific relationships obtained by the ML were only considered in this study.
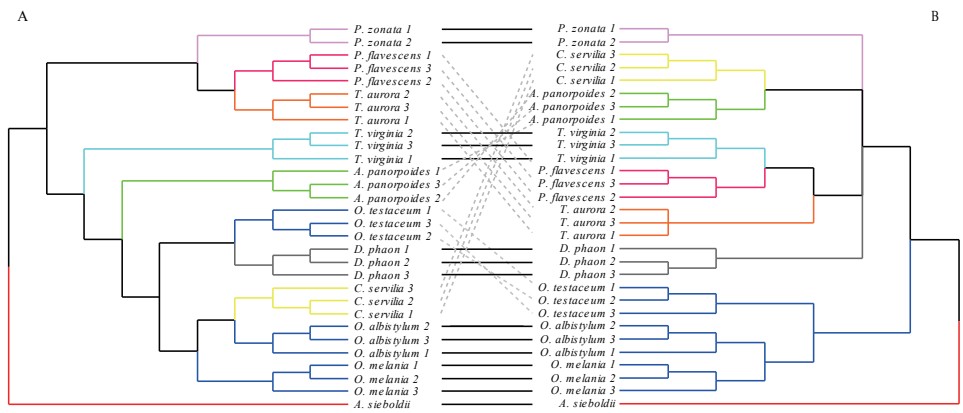

**Figure 7** **The morphological tree of forewings (A) vs. maximum likelihood phylogram obtained from the molecular data (mitochondrial *COI* + nuclear *18S rRNA* + *ITS1* + *5.8S rRNA* + *ITS2* + *28S rRNA* (B).** The clustering of the forewing morphological tree on the left was (—) or was not (…) consistent with the clustering based on the phylogenetic analysis using the molecular data on the right.

## Comparative analysis between the morphological relationship tree of forewings and hindwings obtained based on the UPGMA method and the interspecific relationship tree based on the ML method

The analysis of forewings (Fig. 7) showed that (based on the wing shape) the individuals of each species clustered together first, then clustered with the other species with relatively close morphological relationships. In the morphological tree, the species of genus *Orthetrum* were grouped together, but were mixed with *Crocothemis servilia* and *Deielia phaon*; also, *Pantala flavescens* and *Trithemis aurora* were clustered together. These groupings were consistent with the results of molecular-based genetic analysis. However, for some other species, the results of morphological clustering based on forewings were completely different from those based on the molecular relationships.

The hindwing shape analysis also showed that individuals within the species could be clustered first (Fig. 8). Compared with the results of the forewing shapes, many similarities were found. For example, *Crocothemis servilia* and *Deielia phaon* were also clustered first with *Orthetrum*, *Tramea virginia* was a separate branch, and *Pseudothemis zonata* and *Trithemis aurora* were clustered into a branch. However, the hindwing shape clustering was completely different from that based on the molecular relationships. In general, even though there was some phylogenetic information in the forewing shape, the relationships based on the molecular data were still substantially different. In contrast, there was no interspecific phylogenetic information in the hindwing shape.

## DISCUSSION

### Wing shape and migratory habits

The application of geometric morphometrics method to study wing shape diversity of dragonflies can effectively reveal the relationships among related species (*Outomuro, Dijkstra & Johansson, 2013*; *Klingenberg, 2016*). The PCA results of the forewing shape in

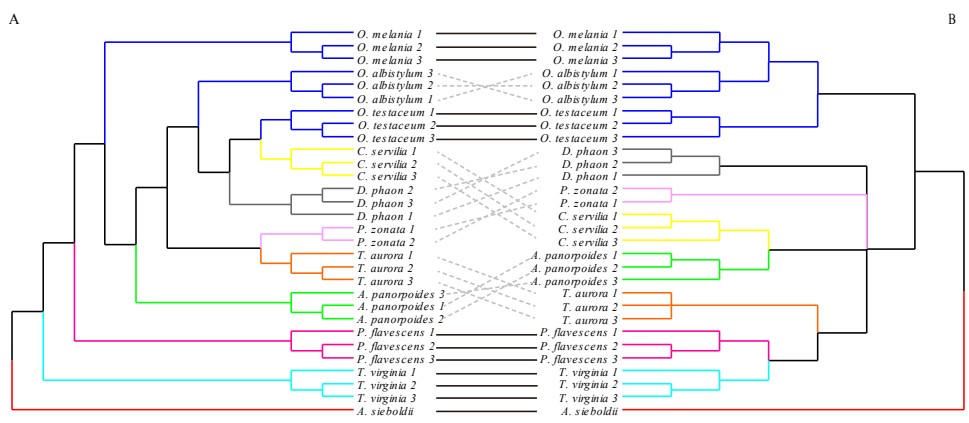

**Figure 8** **The morphological tree of hindwings (A) vs. maximum likelihood phylogram obtained with the molecular dataset (mitochondrial _COI_ + nuclear _18S rRNA_ + _ITS1_ + _5.8S rRNA_ + _ITS2_ + _28S rRNA_ (B).** The clustering of the hindwing morphological tree on the left was (—) or was not (…) consistent with the clustering based on of the phylogenetic analyses using the molecular data on the right.

this study showed the main difference between the 10 species of dragonfly was in the apical and radial regions as well as the discoidal region. In contrast, the main difference in the hindwing shape was in the apical and the anal regions. Based on the dynamic load in flight, the wing nodus provides stability during stroke, with all the wing veins centered on the wing nodus; hence, nodus plays a key role in wing deformation and is likely subjected to extensive bending and torsion in flight (_Rajabi et al., 2017_). The surface of the dragonfly's wings forms various hollow and ridge regions (_Suárez-Tovar & Sarmiento, 2016_), so the wing nodus may be affected by bending as well as twisting deformations during flight. The 10 species of dragonfly exhibit large differences in flight behavior in this study, and these differences in behavior might have led to differences in the wing shape. From the perspective of the wing function, the characteristics of the apical region of dragonfly wing are related to its forward dive and fast flight, playing an important role in long-distance migration, territorial patrol and courtship competition (_Rajabi et al., 2018_). Therefore, the difference in wing shape among different species tested in the present study was expressed prominently at the apical region of the wing. Regardless of the forewing or hindwing, the cubital region and the anal region differed greatly among species. From a functional point of view, these two regions are closely related to the migratory ability of dragonflies (_Sun & Bhushan, 2012_; _Suárez-Tovar & Sarmiento, 2016_ ). It is generally considered that dragonflies with strong migratory ability have larger cubital and anal regions than non-migrating dragonflies (_Jongerius & Lentink, 2010_; _Rajabi et al., 2018_ ).

In this study, the five dragonfly species of _Crocothemis servilia_, _Orthetrum melania_, _Orthetrum albistylum_, _Orthetrum testaceus_ as well as _Acisoma panorpoides_ were distributed mainly on the positive axis of PC1 and PC2. These species had wide and short forewing, with the small anal region of the hindwing. _Tramea virginia_, _Trithemis aurora_, _Pseudothemis zonata_, and _Pantala flavescens_ were distributed mainly on the negative axis of PC1 and PC2. Their forewings were long and narrow, and the anal region of the hindwing was

large. According to the research by *Rajabi et al. (2018)*, the species of dragonfly with long and narrow wing were more suitable for migration, whereas those with wide and short wings were more suited to standing still. Among the dragonflies tested in the present study, from the behavioral point of view, *Tramea virginia*, *Trithemis aurora*, *Pseudothemis zonata*, and *Pantala flavescens* were all species with strong flying ability, conducting stagnation flight and territory patrols, whereas the species *Crocothemis servilia*, *Orthetrum melania*, *Orthetrum albistylum*, *Orthetrum testaceus* and *Acisoma panorpoides* would prefer hovering around ponds or standing still for long periods. The results of this study were in good agreement with those of *Rajabi et al. (2018)*, further confirming the relationship between wing shape and migration.

## Genus, subfamily and family relationships

This study illustrated the preliminarily relationships among species, genera, subfamilies, and families based on the phylogenetic relationships of 10 species of dragonfly based on the mitochondrial *COI* gene and the nuclear genes *18S*, *28S rRNA* and *ITS*. *Deielia phaon* and *Pantala flavescens* showed a close relationship, even though they belong to different subfamilies; moreover, they formed a paraphyletic group with *Acisoma panorpoides* and *Crocothemis servillia*, belonging to subfamily Sympetrinae. This result is similar to the results of *Ware, May & Kjer (2007))* based on the nuclear genes *16S* and *28S rRNA*, and also similar to the results of *Kim et al. (2014)* based on the mitochondrial *COI* gene and the nuclear genes *16S* and *28S rRNA*. In the phylogenetic tree, *Pantala flavescens* and *Trithemis aurora* formed a paraphyletic group, indicating a close relationship despite belonging to different subfamilies; this result was similar to those of *Ware, May & Kjer (2007))*. In this study, the three species of *Orthetrum* were independent as a branch, and far away from other species. In general, the results of this study were consistent with the results of *Kim et al. (2014))*, *Carle, Kjer & May (2015))* and *Yong et al. (2016))*, indicating that the interspecific phylogenetic relationships based on the mitochondrial *COI* gene and the nuclear genes *18S*, *28S rRNA* and *ITS* were reliable, and that these genes can be used as barcode genes for interspecies classification.

## Evaluation of genetic information on wing shape

This study constructed the interspecific relationship trees for the morphological information on forewings and hindwings based on the UPGMA method, and compared them with the phylogenetic trees obtained based on the molecular data by the BI method. The establishment of interspecific relationships using the UPGMA method in morphological analysis can be supported by numerous studies (*Ramírez-Sánchez, Luna & Cramer, 2016*; *Fouquet et al., 2012*; *Gvoždík, Moravec & Kratochvíl, 2008*; *Limsopatham et al., 2018*). The UPGMA method is effective in interspecies morphological analysis, although *Robinson & Terhune (2017)* suggested the UPGMA method in subspecies analysis, such as morphological relationship between subspecies or geographical populations, might obscure the patterns among individuals by the interobserver and intermethod errors. However, in the interspecific relationship analysis, this method is ideal. Using the UPGMA method in the present study to analyze forewing and hindwing shapes, the individuals of

each species were clustered initially into one branch. Then, topological relationships were established with other species. Compared to the intra-species relationships, the interspecific morphological relationship was farther, so they clustered later.

Comparing the morphological relationship tree based on the wing shape and the phylogenetic tree based on the molecular data, some relationships, but also many differences, were found. Regarding the forewing shape, the three species of *Orthetrum* were clustered into a branch, but had *Crocothemis servilia* and *Deielia phaon* mixed in. The phylogenetic tree based on molecular data showed that *Crocothemis servilia* and *Deielia phaon* (subfamily Sympetrinae) had a distant relationship with *Orthetrum* (subfamily Libellulinae). However, from the behavioral point of view, *Crocothemis servilia*, *Deielia phaon* and the species of *Orthetrum* have many similarities, generally living around ponds or streams, resting on grasses and dead branches, or hovering over grass and ponds. Their territorial consciousness is weak and they can coexist with other species. Similar behaviors and habits may be associated with similar forewing and hindwing shapes. In terms of forewing morphology, *Pantala flavescens*, *Trithemis aurora* and *Pseudothemis zonata* were clustered together as one branch, but could not be combined into one branch based on hindwings. From the behavioral point of view, these dragonflies have strong migrating ability that might have influenced clustering based on the morphology. In terms of the molecular data, *Pantala flavescens* and *Trithemis aurora* were clustered together as a branch, and were distant from *Pseudothemis zonata*. These findings showed there was some genetic information in the wing shape, but it was influenced more by the behavior and life habits. Hence, for dragonflies, establishing inter-species relationships based directly on wing shape may be unreliable. *Pilgrim & Vondohlen (2008)* studied the phylogenetic relationships of Sympetrinae based on molecular data (mitochondrial loci 16S and 12S rRNA) and morphological traits (38 wing venation characters); even though the study did not involve direct comparison of phylogenetic trees based on the two types of information, its conclusion was that the characteristics of the wing veins might be useless in the analysis of relationships due to homoplasy. However, morphological and genetic structure may undergo synchronous evolution in other insects, such as the pronotum and genital segments of grasshopper genus *Zoniopoda* (*Pocco et al., 2018*), and the wing veins and genital segments of Euptychiina butterflies (*Marín et al., 2017*), indicating that phylogenetic information may be contained in morphological features of some insects.

Because the sample size selected in this study was relatively small and limited to Libellulidae, the results need to be confirmed on a larger and more diverse collection of species. In the future and using a larger sample size, additional morphological features (such as genital segments) need to be examined to achieve a deeper understanding of the relevance among dragonfly interspecific phylogenetic relationships, morphological evolution and genetic differentiation.

## CONCLUSIONS

In this article, we analyzed the wing morphology and migration status, and compared the molecular and morphology-based phylogenies of Libellulidae. The main results are summarized as follows:

(1) The dragonfly wing shape characteristics are closely related to its migration ability. Species with strong ability to migrate have the forewing shape that is longer and narrower, and have larger anal region, whereas the species that prefer short-distance hovering or standing still for a long time have forewings that are wider and shorter, and the anal region is smaller.

(2) Integrating morphological and molecular data to evaluate the relationship among dragonfly species shows there is some interspecific phylogenetic information in the forewing shape. In the morphological tree, the species of genus *Orthetrum* were grouped together; also, *Pantala flavescens* and *Trithemis aurora* were clustered together. These groupings were consistent with the results of molecular-based genetic analysis. However, hindwing shape had almost no interspecific phylogenetic information.

(3) The forewing and hindwing of dragonflies exhibit an inconsistent pattern of morphological changes in different species, For example, *Trithemis aurora*, *Orthetrum melania* and *Tramea virginia* showed large differences in wings shape and centroid size analysis, which may be due to the different functions of forewings and hindwings in flight and to complex behavior of dragonflies.

## ACKNOWLEDGEMENTS

We thank our institutions for providing infrastructure and other support. We wish to thank Dr. Chao Song for comments and suggestions on DNA sequence analyses and phylogenetic tree construction.

### Funding
This research was supported by the National Natural Science Foundation of China (No. 31402006) and the Zhejiang Provincial Natural Science Foundation of China (No. LY17C040001, No. LGN18C190007). In addition, this work was supported by Taizhou science and technology project (Grant No. 1901ny09). There was no additional external funding received for this study. The funders had no role in study design, data collection and analysis, decision to publish, or preparation of the manuscript.

### Grant Disclosures
The following grant information was disclosed by the authors:
National Natural Science Foundation of China: No. 31402006.
Zhejiang Provincial Natural Science Foundation of China: No. LY17C040001, No. LGN18C190007.
Taizhou science and technology project: 1901ny09.

### Competing Interests
The authors declare there are no competing interests.

## Author Contributions

- Shu-Ting Huang and Hai-Rui Wang performed the experiments, prepared figures and/or tables, authored or reviewed drafts of the paper, and approved the final draft.
- Wan-Qin Yang and Yi Bai conceived and designed the experiments, analyzed the data, prepared figures and/or tables, authored or reviewed drafts of the paper, and approved the final draft.
- Ya-Chu Si, Yu-Tian Wang and Meng-Lian Sun performed the experiments, prepared figures and/or tables, and approved the final draft.
- Xin Qi analyzed the data, prepared figures and/or tables, authored or reviewed drafts of the paper, and approved the final draft.

## DNA Deposition

The following information was supplied regarding the deposition of DNA sequences:

Table 3 contains accession numbers for the mitochondria COI and Nuclear 18S rRNA+ITs1+5.8s+ITs2+28S rRNA sequences.

## Data Availability

The raw measurements data are available in the Supplementary Files.

## Supplemental Information

Supplemental information for this article can be found online at http://dx.doi.org/10.7717/peerj.8567#supplemental-information.

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
