# Peer review of "Phylogeny of Libellulidae (Odonata: Anisoptera): comparison of molecular and morphology-based phylogenies based on wing morphology and migration"

_PeerJ, doi:10.7717/peerj.8567_

## Round 0.1 · original submission · Minor Revisions

Both reviewers were enthusiastic about you study, but also identified a number of minor issues that need to be addressed.

·

Basic reporting

It was my pleasure the review this article and I congratulated the authors on an interesting piece of work. The manuscript is well written and clearly structured. The use of English seems professional throughout, with a few exceptions (see below).
The background provided helped me, as a non-entomologist, understand both the research objectives and the results obtained. The raw data are shared as supplemental information.
Altogether, the article is self-contained and convincingly conveys its message.

I have a few linguistic issues to consider:

line 32: Paup misspelled
lines 71-74: reading these two sentences, I have difficulties understanding the difference between the two views. Please rephrase.
lines 81-84: what do you mean by "population evolution law" and "systematic generation relationship"? Please explain or use other expressions here.
line 92: delete "study"
lines 102-103: what is the "law of genetic difference"? I am not familiar with this expression.
line 109: is "coevolution" the right word to use here? I think you need to reformulate your ideas here.
line 129: check spacing at brackets
line 130: delete “its”
line 139: verb missing?
lines 139-140: “Dragonfly” is not a species at all. Please rephrase, either by mentioning the actual species you refer to or by referring to dragonflies as a clade (specify the relevant taxonomic rank).
Caption of Fig. 1: check spacing
line 299: “hindwing” instead of “findwing”
line 316: “dragonfly species” instead of “species dragonflies”
lines 385-386: “due to homoplasy” instead of “due to the trait homoplasy”
line 392: replace “so” by a comma
Caption of Fig. 7: “The clustering of the forewing…” instead of “he clustering of the forewing…”
Acknowledgments: this section includes a funding statement. As far as I understand, this is not the right place for funding statements according to journal guidelines.

Experimental design

The research presented seems novel to me and falls within the scope of the journal. It addresses an unambiguously defined aim that seems both relevant and meaningful to me. The match or mismatch between molecular phylogenies and morphological characters is a highly interesting but still poorly understood topic. The more rigorously performed studies, such as the present one, the better our knowledge.
For some of the methods described, I lack the necessary expertise to judge the validity of the details provided in full extent. The phylogenetic and morphometric approaches, however, seem sound to me.

Validity of the findings

The findings presented seem valid to me, well-presented and supported by the data provided. I cannot assess the statistical soundness of all data so it would be useful to let someone with the relevant expertise review this aspect specifically.
The conclusions presented refer to the research questions raised and seem perfectly sound to me.

Additional comments

Good job! Looking forward to seeing this published.

·

Basic reporting

Are the interspecific relationships established based on genetic information consistent with the traditional classification of Libellulidae dragonflies which is based on the wing shape?
This is an interesting question, which the authors, aimed to answer that by combining geometric morphometrics method, principal component analysis (PCA), cluster analysis (UPGMA), etc.
The manuscript is well written, introduction is well motivated, methods are suitable, and results are well presented.

Experimental design

The only drawback is small sample size (the authors focused on only 10 species from a single family), which to some extent narrows the scope of the manuscript and raises the question whether the results can be generalized to other species and families or not. However, it still facilitates more comprehensive future studies.

Validity of the findings

All criteria were met.

Additional comments

I would like to suggest the acceptance of the paper with following minor comments.
1. Something that I was missing is the images of the wings analyzed in this study in comparable size, which can be added as suppl mat.
2. Line 116
this sentence would benefit from a more relevant citation.
3. Line 139
needs revision.
4. Line 279
Based on
5. Line 298
for a non-specialist, it would be nice to denote “apical”, “radial”, etc. in a wing image.
6. Line 299 and line 300 “dominated by the wing nodus”
What does this exactly mean?
7. Line 299 “findwing”
8. Line 301 “is the basis of the whole wing structure”
It is a vague expression.
9. Line 302 “wing nodus is the main load-bearing”
Rajabi et al, in their paper, indicated that the nodus plays a key role in wing deformation and likely subjected to extensive bending and torsion in flight.
10. Line 313-315
These sentences require citation.

Hamed Rajabi, D. Sc. D. Eng.

---

## Round 0.2 · Minor Revisions

I believe that you did a good job in terms of the scientific contact. However, the style and grammar of the text still need work in a revision round. I understand that you already payed a company to review the text, but unfortunately that was not sufficient.

---

## Round 0.3 · accepted · Accept

I am satisfied with the latest version of the text.